# On Convergence of Average-Reward Off-Policy Control Algorithms in Weakly-Communicating MDPs

## Abstract

We show two average-reward off-policy control algorithms, Differential Q Learning (Wan, Naik, & Sutton 2021a) and RVI Q Learning (Abounadi Bertsekas & Borkar 2001), converge in weakly-communicating MDPs. Weakly-communicating MDPs are the most general class of MDPs that a learning algorithm with a single stream of experience can guarantee obtaining a policy achieving optimal reward rate. The original convergence proofs of the two algorithms require that all optimal policies induce unichains, which is not necessarily true for weakly-communicating MDPs. To the best of our knowledge, our results are the first showing average-reward off-policy control algorithms converge in weakly-communicating MDPs. As a direct extension, we show that average-reward options algorithms introduced by (Wan, Naik, & Sutton 2021b) converge if the Semi-MDP induced by options is weakly-communicating.

## 1 Introduction

Modern reinforcement learning algorithms are designed to maximize the agent's goal in either the episodic setting or the continuing setting. In both settings, there is an agent continually interacting with its world, which is usually assumed to be a Markov Decision Process (MDP). In episodic problems, there is a special terminal state and a set of start states. If the agent reaches the terminal state, it will be reset to one of the start states. Continuing problems are different in that there is no terminal state, and the agent will never be reset by the world. For continuing problems, two commonly considered objectives are the discounted objective and the average-reward objective. The discount factor in the discounted objective has been observed to be deprecated in the function approximation control setting, suggesting that the average-reward objective might be more suitable for continuing problems.

In this paper, we consider off-policy control algorithms for the average-reward objective. These algorithms learn a policy that achieves the best possible average-reward rate, using data generated by some other policy that the agent may not have control of. Designing convergent off-policy algorithms for the average-reward objective is challenging. While there are several off-policy learning algorithms in the literature, the only known convergent algorithms are SSP Q-learning and RVI Q-learning, both by Abounadi, Bertsekas, & Borkar (2001), the algorithm by Ren & Krogh (2001), and Differential Q-learning by Wan, Naik, & Sutton (2021a). Others either do not have convergence proofs (Schwartz 1993; Singh 1994; Bertsekas & Tsitsiklis 1996; Das 1999), or have incorrect proofs (Yang 2016; Gosavi 2004). [1]

The algorithm by Ren & Krogh (2001) requires knowledge of properties of the MDP which are not typically known. The convergence of SSP Q-learning is limited in MDPs with a state being recurrent under all policies. The convergence of the RVI Q-learning algorithm (Abounadi et al. 2001) was developed for unichain MDPs, which just means that the Markov chain induced by any stationary policy is unichain [2]. The convergence of Differential Q-learning (Wan et al. 2021a) requires a weaker

---

[1] See Appendix D in Wan et al. (2021a) for a discussion about Yang's proof and see Appendix C of this paper for a discussion about Gosavi's proof.

[2] A Markov chain is unichain if there is only one recurrent class in Markov chain, plus a possibly empty set of transient states.

assumption – all optimal policies being unichain. It is clear that RVI Q-learning also converges under this assumption with a small modification of its proof. It is not rare that an optimal policy induces multiple recurrent classes. For example, consider the MDP at the bottom of Figure 1. If an optimal policy induces multiple recurrent classes, the proofs of RVI Q-learning and Differential Q-learning would not go through. Technically, this is because both two proofs require that the uniqueness of the solution for the action-value function up to an additive constant in the average-reward optimality equation.

A more general class of MDPs, called weakly-communicating MDPs, is not limited in the number of recurrent classes under the optimal policy. In these MDPs, all policies may induce multiple recurrent classes. The only assumption is that, except for a set of states that are transient under every policy, all states are reachable from every other state in a finite number of steps with a non-zero probability. It has been observed that the set of weakly-communicating MDPs is the most general set of MDPs such that there exists a learning algorithm that can, using a single stream of experience, guarantee to identify a policy that achieves the optimal average reward rate in the MDP (Barlett & Tewari 2009).

In this paper, we show convergence of RVI Q-learning and Differential Q-learning in weakly-communicating MDPs, without requiring any additional assumptions compared with their original convergence results. Two key steps in our proof are 1) showing that the solution sets of the two algorithms are non-empty, closed, bounded, and connected, and 2) showing that 0 is the unique solution for the Bellman optimality equation when all rewards are 0. With these two results, we use asynchronous stochastic approximation results by Borkar (2009) to show convergence to the solution sets. As a direct extension of the above results, we also show the convergence of two algorithms that extend the Differential Q-learning algorithm to the options framework, introduced by Wan et al. (2021b), if the Semi-MDP induced by the MDP and the set of options is weakly-communicating.

## 2  PRELIMINARIES

Consider a finite Markov decision process, defined by the tuple $\mathcal{M} \doteq (\mathcal{S}, \mathcal{A}, \mathcal{R}, p)$, where $\mathcal{S}$ is a set of states, $\mathcal{A}$ is a set of actions, $\mathcal{R}$ is a set of rewards, and $p : \mathcal{S} \times \mathcal{R} \times \mathcal{S} \times \mathcal{A} \to [0, 1]$ is the dynamics of the environment. Every time steps $t$ the agent observes the state of the MDP $S_t \in \mathcal{S}$ and chooses an action $A_t \in \mathcal{A}$ using some policy $b : \mathcal{A} \times \mathcal{S} \to [0, 1]$, then receives from the environment a reward $R_{t+1} \in \mathcal{R}$ and the next state $S_{t+1} \in \mathcal{S}$, and so on. The transition dynamics are defined as $p(s', r \mid s, a) \doteq \Pr(S_{t+1} = s', R_{t+1} = r \mid S_t = s, A_t = a)$ for all $s, s' \in \mathcal{S}, a \in \mathcal{A}$, and $r \in \mathcal{R}$. Denote the set of stationary Markov policies $\Pi$.

The reward rate of a policy $\pi$ starting from a given start state $s$ can be defined as:

$$r(\pi, s) \doteq \lim_{n \to \infty} \frac{1}{n} \sum_{t=1}^{n} \mathbb{E}[R_t \mid S_0 = s, A_{0:t-1} \sim \pi]. \quad (1)$$

Given an arbitrary MDP, the agent may not even be able to visit all states and would therefore miss the chance of learning, for every state $s$, a policy that achieves the optimal reward rate $\sup_\pi r(\pi, s)$ and the agent can at most learn an optimal policy for a set of states, in which every state is reachable from every other state. Such a set of states is often called *communicating*. Formally speaking, we say a set of states *communicating*, if there exists a policy such that moving from either one state in the set to the other one in the set in a finite number of steps has positive probability. If the entire state space is communicating, we say an MDP *communicating*. *Weakly-communicating*

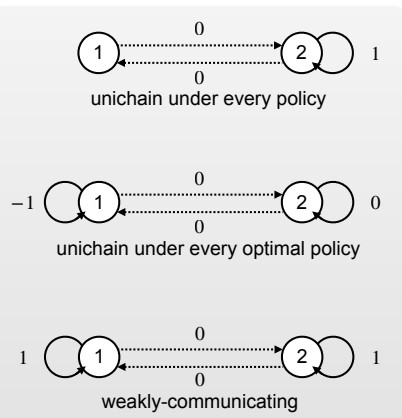

Figure 1: Examples of three different types of MDPs. In each of the three MDPs, there are two states marked by two circles respectively. There are two actions *solid* and *dashed*, both causing deterministic effects. *Top*: The MDP is unichain under every stationary policy. *Middle*: There are two deterministic optimal policies, moving to state 2 or going back and forth between state 1 and state 2. Other optimal stationary policies are mixtures of these two deterministic policies. All optimal policies are unichain. *Down*: There are three deterministic optimal policies: moving to state 1, moving to state 2, and staying at the current state. Other optimal stationary policies are mixtures of these two deterministic policies. The policy of staying at the current state induces two recurrent classes.

MDPs generalizes over communicating MDPs. In weakly-communicating MDPs, in addition to a *closed* communicating set of states, there is a possibly empty set of states that are under every policy.

For weakly-communicating MDPs, there exists a unique optimal reward rate $r_*$, which does not depend on the start state. We say a policy is optimal if it achieves $r_*$ regardless of the start state. The goal of an off-policy control algorithm is to learn an optimal policy from the stream of experience $\dots, S_t, A_t, R_{t+1}, S_{t+1}, \dots$ generated by a behavior policy that is not necessarily the same as the agent's learned policy. Both RVI Q-learning and Differential Q-learning achieve this goal by solving $\bar{r}$ and $q$ in the optimality equation:

$$q(s,a) = \sum_{s',r} p(s',r \mid s,a)(r - \bar{r} + \max_{a'} q(s',a')), \quad \forall\, s \in \mathcal{S}, a \in \mathcal{A}. \tag{2}$$

It is known that $r_*$ is the unique solution of $\bar{r}$ and any greedy policy w.r.t. any solution of $q$ is an optimal policy. In addition, shifting any solution of $q$ by any constant vector results in the other solution of $q$. Finally, unlike in unichain MDPs (or MDPs where all optimal policies are unichain) where all solutions are different by some constant vector, in weakly-communicating MDPs, solutions of $q$ may have multiple degrees of freedom. That is, if $q_1, q_2$ are both solutions of $q$, it is possible that $q_1 \neq q_2 + ce, \forall\, c \in \mathbb{R}$, where $e$ denotes the all-one vector.

If the agent has a set of options, it may choose to execute these options. Each option $o$ in $\mathcal{O}$ has two components: the option's *policy* $\pi^o : \mathcal{A} \times \mathcal{S} \to [0,1]$, and the termination probability $\beta^o : \mathcal{S} \to [0,1]$. For simplicity, for any $s \in \mathcal{S}, o \in \mathcal{O}$, we use $\pi(a \mid s, o)$ to denote $\pi^o(a,s)$ and $\beta(s,o)$ to denote $\beta^o(s)$. If the agent executes option $o$ at state $s$, the option's policy is followed, until the option terminates. Let $\mathcal{L}$ be the set of all possible lengths of options and $\hat{\mathcal{R}}$ be the set of all possible cumulative rewards. Note that $\mathcal{L}$ and $\hat{\mathcal{R}}$ are possibly countably infinite. Let $\hat{p}(s',r,l \mid s,o)$ be, when executing option $o$ starting from state $s$, the probability of terminating at state $s'$, with cumulative reward $r$ and length $l$. Formally, for any $s, s' \in \mathcal{S}, o \in \mathcal{O}, r \in \hat{\mathcal{R}}, l \in \mathcal{L}$, $\hat{p}$ can be defined recursively in the following way:

$$\hat{p}(s',r,l \mid s,o) = \sum_a \pi(a \mid s,o) \sum_{\tilde{s},\tilde{r}} p(\tilde{s},\tilde{r} \mid s,a)$$

$$[\beta(\tilde{s},o)\mathbf{I}(\tilde{s}=s',\tilde{r}=r,\tilde{l}=1) + (1-\beta(\tilde{s},o))\hat{p}(s',r-\tilde{r},l-1 \mid \tilde{s},o)], \tag{3}$$

where $\mathbf{I}$ is an indicator function.

An MDP $\mathcal{M}$ and a set of options $\mathcal{O}$ results in an Semi-MDP (SMDP) $\hat{\mathcal{M}} = (\mathcal{S}, \mathcal{O}, \mathcal{L}, \hat{\mathcal{R}}, \hat{p})$.

If the agent chooses options using a meta-policy (policy-over-options) $\pi : \mathcal{O} \times \mathcal{S} \to [0,1]$ and execute these options, we denote $\dots, \hat{S}_n, \hat{O}_n, \hat{R}_{n+1}, \hat{S}_{n+1}, \dots$ as the sequence of option transitions. For this SMDP, the *reward rate* of $\pi$ given a start state $s$ can be defined as $r^C(\pi,s) \doteq \lim_{t\to\infty} \mathbb{E}_\pi[\sum_{i=1}^t R_i \mid S_0 = s]/t$. $r(\pi,s) \doteq \lim_{n\to\infty} \mathbb{E}_\pi[\sum_{i=0}^n \hat{R}_i \mid \hat{S}_0 = s]/\mathbb{E}_\pi[\sum_{i=0}^n \hat{L}_i \mid \hat{S}_0 = s]$. Both limits exist and are equivalent (Puterman's (1994) propositions 11.4.1) under the following assumption:

**Assumption 1.** For each option $o \in \mathcal{O}$, when executing the option, there is a non-zero probability of terminating the option after at most $|\mathcal{S}|$ stages, regardless of the initial states.

**Proposition 1.** *Under Assumption 1, the expected value as well as the variance of the execution time and cumulative reward of every option at every state exist and are finite.*

We say an SMDP is weakly-communicating if the MDP with state space $\mathcal{S}$, action space $\mathcal{O}$, reward space $\hat{\mathcal{R}}$, and transition function $\sum_l \hat{p}(s,r,l \mid s,o)$ is weakly-communicating. Just like the MDP setting, if the SMDP is weakly-communicating, the optimal reward rate $\hat{r}_* \doteq \sup_{\pi \in \hat{\Pi}} r(\pi,s)$, where $\hat{\Pi}$ denotes the set of stationary Markov meta-policies, does not depend on the start state $s$. In addition, the solutions of $q$ may not be different for a constant. Given an MDP and a set of options, the goal of the off-policy control problem is to find a policy that achieves $\hat{r}_*$. Inter-option Differential Q-learning achieves this goal by solving the *optimality* equation for SMDPs (Puterman 1994):

$$q(s,o) = \sum_{s',r,l} \hat{p}(s',r,l \mid s,o)\big(r - \bar{r}\cdot l + \max_{o'} q(s',o')\big), \tag{4}$$

where $q$ and $\bar{r}$ denote estimates of the option-value function and the reward rate respectively. Just like the MDP setting, $\bar{r}$ has $\hat{r}_*$ as its unique solution and solutions of $q$ may not be different by a constant vector.

Intra-option Differential-learning finds an optimal policy by solving the *intra-option optimality equation*.

$$q(s,o) = \sum_a \pi(a \mid s,o) \sum_{s',r} p(s',r \mid s,a)(r - \bar{r} + u_*^q(s',o)), \quad \forall s \in \mathcal{S}, o \in \mathcal{O}, \quad (5)$$

where

$$u_*^q(s',o) \doteq \big(1 - \beta(s',o)\big)q(s',o) + \beta(s',o)\max_{o'} q(s',o'). \quad (6)$$

The following proposition shows that the set of solutions of equation 5 is the same with that of equation 4.

**Proposition 2.** *Any solution of equation 4 is also a solution of equation 5 and vice versa.*

## 3 CONVERGENCE RESULTS

In this section, we present the convergence theory of Differential Q-Learning, RVI Q-Learning in weakly-communicating MDPs and that of the two option extensions of Differential Q-Learning in weakly-communicating SMDPs. Empirical results verifying the convergence of the two MDP algorithms are presented in Appendix B.

Differential Q-learning updates a table of estimates $Q_t : \mathcal{S} \times \mathcal{A} \to \mathbb{R}$ as follows:

$$Q_{t+1}(S_t, A_t) \doteq Q_t(S_t, A_t) + \alpha_{\nu(t,S_t,A_t)}\delta_t, \quad (7)$$
$$Q_{t+1}(s,a) \doteq Q_t(s,a), \ \forall s, a \neq S_t, A_t,$$

where $\nu(t, S_t, A_t)$ is the number of times $S_t, A_t$ has been visited before time step $t$, $\alpha_{\nu(t,S_t,A_t)}$ is a step-size sequence, and $\delta_t$, the temporal-difference (TD) error, is:

$$\delta_t \doteq R_{t+1} - \bar{R}_t + \max_a Q_t(S_{t+1}, a) - Q_t(S_t, A_t), \quad (8)$$

where $\bar{R}_t$ is a scalar estimate of $r_*$, updated by:

$$\bar{R}_{t+1} \doteq \bar{R}_t + \eta\alpha_{\nu(t,S_t,A_t)}\delta_t, \quad (9)$$

and $\eta$ is a positive constant.

We now present the convergence theorem of Differential Q-learning. We first state the required assumptions, which are also required by the original convergence proof of Differential Q-Learning by Wan et al. (2021a).

**Assumption 2.** For all $n \geq 0$, $\alpha_n > 0$, $\sum_{n=0}^{\infty} \alpha_n = \infty$, and $\sum_{n=0}^{\infty} \alpha_n^2 < \infty$.

**Assumption 3.** Let $[\cdot]$ denote the integer part of $(\cdot)$, for $x \in (0,1)$, $\sup_n \frac{\alpha_{[xn]}}{\alpha_n} < \infty$ and $\frac{\sum_{n=0}^{[ym]} \alpha_n}{\sum_{n=0}^{m} \alpha_n} \to 1$ uniformly in $y \in [x,1]$.

**Assumption 4.** There exists $\Delta > 0$ such that

$$\liminf_{n \to \infty} \frac{\nu(n,s,a)}{n+1} \geq \Delta,$$

a.s., for all $s \in \mathcal{S}, a \in \mathcal{A}$. Furthermore, for all $x > 0$, let $N(n,x) = \min\left\{m > n : \sum_{k=n+1}^{m} \alpha_k \geq x\right\}$, the limit $\lim_{n\to\infty}\left(\sum_{k=\nu(n,s,a)}^{\nu(N(n,x),s,a)} \alpha_k\right) / \left(\sum_{k=\nu(n,s',a')}^{\nu(N(n,x),s',a')} \alpha_k\right)$ exists a.s. for all $s, s' \in \mathcal{S}, a, a' \in \mathcal{A}$.

**Theorem 1.** *If $\mathcal{M}$ is communicating and Assumptions 1–4 hold, the Differential Q-learning algorithm (Equations 7–9) converges, almost surely, $\bar{R}_t$ to $r_*$, $Q_t$ to the set of solutions of equation 2 and*

$$r_* - \bar{R}_0 = \eta\left(\sum_{s,a} q(s,a) - \sum_{s,a} Q_0(s,a)\right), \quad (10)$$

*and $r(\pi_t, s)$ to $r_*$, for all $s \in \mathcal{S}$, where $\pi_t$ is any greedy policy w.r.t. $Q_t$.*

**Remark:** If the MDP is weakly-communicating, that is, it contains transient states, the agent eventually reaches the closed communicating state and never returns to the transient states. Elements in $Q_n$ that are associated with the closed communicating set converge to a set that depends on the values of $Q$ and $\bar{R}$ when the MDP reaches the closed communicating set for the first time. Other elements in $Q_n$ would only be visited for a finite number of times and can not be guaranteed to converge to their correct values by any learning algorithms. Other conclusions of the theorem remain unchanged. This observation on weakly-communicating MDPs apply also to Theorems 2–4.

The update rules of RVI Q-Learning are

$$Q_{t+1}(S_t, A_t) \doteq Q_t(S_t, A_t) + \alpha_{\nu(t, S_t, A_t)}\delta_t(S_t, A_t) \tag{11}$$
$$Q_{t+1}(s, a) \doteq Q_t(s, a), \ \forall\, s, a \neq S_t, A_t,$$

where

$$\delta_t(S_t, A_t) \doteq R_{t+1} - f(Q_t) + \max_a Q_t(S_{t+1}, a) - Q_t(S_t, A_t). \tag{12}$$

and $f : \mathcal{S} \times \mathcal{A} \to \mathbb{R}$ satisfying the following assumption.

**Assumption 5.** 1) $f$ is $L$-Lipschitz, 2) there exists a positive scalar $u$ s.t. $f(e) = u$ and $f(x + ce) = f(x) + cu$, 3) $f(cx) = cf(x)$.

**Theorem 2.** *If $\mathcal{M}$ is communicating and Assumptions 1–5 hold, then the RVI Q-Learning algorithm (Equations 11–12) converges, almost surely, $\bar{R}_t$ to $r_*$, $Q_t$ to the set of solutions of equation 2 and*

$$r_* = f(q), \tag{13}$$

*and $r(\pi_t, s)$ to $r_*$, for all $s \in \mathcal{S}$, where $\pi_t$ is any greedy policy w.r.t. $Q_t$.*

Now consider option extensions of Differential Q-Learning. Given an SMDP $\hat{\mathcal{M}} = (\mathcal{S}, \mathcal{O}, \mathcal{L}, \hat{\mathcal{R}}, \hat{p})$, inter-option Differential Q learning maintains estimates of option values, and, inspired by Schweitzer (1971), update estimates using scaled TD errors:

$$Q_{n+1}(\hat{S}_n, \hat{O}_n) \doteq Q_n(\hat{S}_n, \hat{O}_n) + \alpha_{\nu(n, \hat{S}_n, \hat{O}_n)}\delta_n / L_n(\hat{S}_n, \hat{O}_n), \tag{14}$$
$$Q_{n+1}(s, o) \doteq Q_n(s, o), \ \forall\, s, o \neq \hat{S}_n, \hat{O}_n,$$
$$\bar{R}_{n+1} \doteq \bar{R}_n + \eta\alpha_{\nu(n, \hat{S}_n, \hat{O}_n)}\delta_n / L_n(\hat{S}_n, \hat{O}_n), \tag{15}$$

where $\nu(n, \hat{S}_n, \hat{O}_n)$ is the number of visits before stage $n$, $L_n(\cdot, \cdot)$ comes from an additional vector of estimates $L : \mathcal{S} \times \mathcal{O} \to \mathbb{R}$ that approximates the expected lengths of state-option pairs, updated from experience by:

$$L_{n+1}(\hat{S}_n, \hat{O}_n) \doteq L_n(\hat{S}_n, \hat{O}_n) + \beta_{\nu(n, \hat{S}_n, \hat{O}_n)}(\hat{L}_n - L_n(\hat{S}_n, \hat{O}_n)), \tag{16}$$

where $\{\beta_n\}$ is an another step-size sequence. The TD-error $\delta_n$ in equation 14 and equation 15 is

$$\delta_n \doteq \hat{R}_n - L_n(\hat{S}_n, \hat{O}_n)\bar{R}_n + \max_o Q_n(\hat{S}_{n+1}, o) - Q_n(\hat{S}_n, \hat{O}_n), \tag{17}$$

**Theorem 3.** *If $\hat{\mathcal{M}}$ is communicating, Assumptions 1–4 hold, except for using $\nu(n, s, o)$ instead of $\nu(t, s, a)$, and that $0 \leq \beta_n \leq 1$, $\sum_n \beta_n = \infty$, and $\sum_n \beta_n^2 < \infty$, inter-option Differential Q-learning (Equations 14-17) converges, almost surely, $Q_n$ the set of solutions of equation 4 and*

$$\hat{r}_* - \bar{R}_0 = \eta(\sum_{s,o} q(s, o) - \sum_{s,o} Q_0(s, o)), \tag{18}$$

*$\bar{R}_n$ to $\hat{r}_*$, and $r(\pi_n, s)$ to $\hat{r}_*$ where $\pi_n$ is a greedy policy w.r.t. $Q_n$.*

Intra-option Differential Q learning also maintain estimates of option-values. However, instead of updating the estimates using option transitions, it updates for all options using each action transition.

$$Q_{t+1}(S_t, o) \doteq Q_t(S_t, o) + \alpha_{\nu(t, S_t, o)}\rho_t(o)\delta_t(o), \quad \forall\, o \in \mathcal{O}, \tag{19}$$
$$Q_{t+1}(s, o) \doteq Q_t(s, o), \quad \forall\, s \in \mathcal{S}, o \in \mathcal{O},$$
$$\bar{R}_{t+1} \doteq \bar{R}_t + \eta \sum_{o \in \mathcal{O}} \alpha_{\nu(t, S_t, o)}\rho_t(o)\delta_t(o), \tag{20}$$

where $\alpha_t$ is a step-size sequence, $\rho_t(o) \doteq \frac{\pi(A_t | S_t, o)}{\pi(A_t | S_t, O_t)}$ is the importance sampling ratio, and:

$$\delta_t(o) \doteq R_{t+1} - \bar{R}_t + u_*^{Q_t}(S_{t+1}, o) - Q_t(S_t, o), \tag{21}$$

where $u_*$ is defined in equation 6.

**Theorem 4.** *If $\hat{\mathcal{M}}$ is communicating, Assumptions 1–4 hold, except for using $\nu(t, s, o)$ instead of $\nu(t, s, a)$, intra-option Differential Q-learning (Equations 19-21) converges, almost surely, $Q_t$ to the set of solutions of equation 4 and equation 18, $\bar{R}_t$ to $\hat{r}_*$, and $r(\pi_t, s)$ to $\hat{r}_*$ where $\pi_t$ is a greedy policy w.r.t. $Q_t$.*

## 4    CHARACTERIZATION OF THE SOLUTION SET

In this section, we characterize the sets that the algorithms described in the above section converges to. This section plays a key role in showing their convergence.

We consider the set of solutions of the SMDP optimality equation 4 and

$$c = f(q), \tag{22}$$

where $f : \mathcal{S} \times \mathcal{O} \to \mathbb{R}$ satisfies Assumption 5, $c$ is an arbitrary real. Denote this set as $\mathcal{Q}_\infty$. It is clear that equation 4 generalizes equation 2 and equation 22 generalizes equation 10, equation 13, and equation 18. And thus the characterization of $\mathcal{Q}_\infty$ applies to sets that action/option values in the aforementioned algorithms are claimed to converge to in Theorem 1–Theorem 4.

It is known that if the SMDP is weakly-communicating, equation 4 has $\hat{r}_*$ as its unique solution for $\bar{r}$. For $q$, it has been shown by Schweitzer & Federgruen (1978) (we will refer this work for multiple times and thus we use a shorthand 'S&F' for simplicity from now on) in their Theorem 4.2 that the set of solutions for $q$ in equation 4 is closed, unbounded, connected, and possibly non-convex. The next theorem characterizes $\mathcal{Q}_\infty$.

**Theorem 5.** *If the SMDP is weakly-communicating and $f$ 1) is Lipschitz, and 2) there exists some $u \neq 0$ such that $f(x + ce) = f(x) + cu, \forall\, c \in \mathbb{R}$, $\mathcal{Q}_\infty$ is non-empty, closed, bounded, connected, and possibly non-convex.*

Before presenting the proof, first note that our convergence proof does not rely on the convexity property and we delay the proof of non-convexity to Appendix A.5.

*Proof.* First, $\mathcal{Q}_\infty$ is non-empty. To see this point, note that for any solution of $q$ in equation 2 , $q_*$, $q_* + ce$ is also a solution for any $c \in \mathbb{R}$ and thus there must be a $c$ such that equation 22 holds because $f(x + ce) = f(x) + c$ for any $x, c$.

$\mathcal{Q}_\infty$ is closed because the set of solutions of $q$ in equation 4 is closed by S&F, the set of solutions of $q$ in equation 22 is closed because $f$ is Lipschitz, and the intersection of two closed sets is closed.

**Boundedness**

We now show that $\mathcal{Q}_\infty$ is bounded. To this end, it is enough to show that the solution set of $v$ in

$$v(s) = \max_o \sum_{s',r,l} \hat{p}(s', r, l \mid s, o)(r - l\hat{r}_* + v(s')), \forall\, s \in \mathcal{S} \tag{23}$$

$$c = f(\tilde{r} + \tilde{P}v), \tag{24}$$

where $c$ is any constant, and

$$\tilde{r}(s, o) \doteq \sum_{s',r,l} \hat{p}(s', r, l \mid s, o)(r - l\hat{r}_*)$$

$$\tilde{P}(s, o, s') \doteq \sum_{r,l} \hat{p}(s', r, l \mid s, o)$$

is bounded. Denote this set as $\mathcal{V}_\infty$. Once we show $\mathcal{V}_\infty$ is bounded, $\mathcal{Q}_\infty$ is also bounded because each solution for $q$ can be obtained from a solution for $v$ with a linear operation:

$$q(s, o) = \sum_{s',r,l} \hat{p}(s', r, l \mid s, o)(r - l\bar{r} + v(s')), \quad \forall\, s \in \mathcal{S}, o \in \mathcal{O}. \tag{25}$$

In order to show boundedness, we need the following two lemmas, which slightly modify Theorem 4.1 and 5.1 by S&F. To this end, we need to first introduce some definitions.

For any $\pi \in \hat{\Pi}$, $P_\pi$ denotes the $|\mathcal{S}| \times |\mathcal{S}|$ transition probability matrix under policy $\pi$. That is,

$$P_\pi(s, s') \doteq \sum_{o,r,l} \pi(o \mid s) \hat{p}(s', r, l \mid s, o). \tag{26}$$

Define $P_\pi^\infty$ to be the *limiting matrix* of $P_\pi$, which is the Cesaro limit of the sequence $\{P_\pi^i\}_{i=1}^\infty$:

$$P_\pi^\infty \doteq \lim_{n \to \infty} \frac{1}{n} \sum_{i=0}^{n-1} P_\pi^i. \tag{27}$$

Because $\mathcal{S}$ is finite, the Cesaro limit exists and $P_\pi^\infty$ is a stochastic matrix (has row sums equal to 1).

Let $l_\pi(s) \doteq \sum_{o,s',r,l} \pi(o \mid s) \hat{p}(s', r, l \mid s, o) l$. And let the fundamental matrix $Z_\pi \doteq (I - P_\pi^\infty + P_\pi^\infty)^{-1} = I + \lim_{\gamma \uparrow 1} \sum_{n=1}^\infty \gamma^n (P_\pi^n - P_\pi^\infty)$. Let $b(v, \pi)_i \doteq [r_\pi - l_\pi \hat{r}_* + P_\pi^\infty v - v]_i$ and $\hat{\Pi}_*$ denote the set of optimal meta-policies.

The following two lemmas are the similar to Theorem 4.1 (c) and Theorem 5.1 in S&F, except that 1) the 'max' operates over the set of all optimal policies, instead of the set of all deterministic optimal policies as in Theorem 4.1 (c), and 2) we consider the set of weakly-communicating SMDPs while S&F considers general multi-chain SMDPs. The proofs are also essentially the same. For completeness, we provide the proofs for these two lemmas in Sections A.3–A.4.

**Lemma 1.** *If the SMDP is weakly-communicating, $v$ is a solution of equation 23 if and only if*

$$v(s) = \max_{\pi \in \hat{\Pi}_*} [Z_\pi(r_\pi - l_\pi \hat{r}_*) + P_\pi^\infty v](s), \forall\, s \in \mathcal{S}, \tag{28}$$

*where $\hat{\Pi}_*$ denotes the set of optimal meta-policies.*

Define $R_\pi$ as the set of recurrent states for $P_\pi^\infty$ (i.e., $R_\pi \doteq \{s \mid P_\pi^\infty(s, s) > 0\}$). Let $R_* \doteq \{s \mid s \in R_\pi \text{ for some policy } \pi \in \hat{\Pi}_*\}$. S&F shows that there exists a policy $\pi \in \hat{\Pi}_*$ such that $R_\pi = R_*$. Let $n(\pi)$ be the number of recurrent classes for $P_\pi^\infty$ and let $n_* \doteq n(\pi)$. Denote the set of recurrent classes of $P_\pi$ as $\{R_{*\alpha} \mid 1, 2, \cdots, n_*\}$. The following lemma shows that the solution set of equation 23 has $n_*$ degrees of freedom.

**Lemma 2.** *If the SMDP is weakly-communicating, suppose $v$ and $v + x$ are both solutions of equation 23, then there exists $n_*$ constants $y_1, y_2, \cdots, y_{n_*}$ such that*

$$x(s) = y_\alpha, i \in R_{*\alpha}, \quad \alpha = 1, \ldots, n_* \tag{29}$$

$$x(s) = \max_{\pi \in \hat{\Pi}_*} Z_\pi b(v, \pi)_s + \sum_{\beta=1}^{n_*} \left( \sum_{s' \in R_{*\beta}} P_\pi^\infty(s, s') \right) y_\beta, \quad s \in \mathcal{S} \backslash R_*, \tag{30}$$

$$y_\alpha \geq Z_\pi b(v, \pi)_s + \sum_{\beta=1}^{n_*} \left( \sum_{s' \in R_{*\beta}} P_\pi^\infty(s, s') \right) y_\beta, \quad \alpha = 1, \ldots, n_*, s \in R_{*\alpha}, \pi \in \hat{\Pi}_*. \tag{31}$$

For any $\beta \in \{1, 2, \cdots, n_*\}$, note that there must exist a policy $\pi(\beta)$ such that $R_{*\beta}$ is the only one recurrent class under $\pi(\beta)$. To see this point, note that the SMDP is weakly-communicating, and thus we can modify $\pi$ to obtain a new policy such that all states except for those in $R_{*\beta}$ are transient. Such a policy satisfies our requirement.

Applying this observation and Lemma 2, for any $v \in \mathcal{V}$, we have for any given $\beta \in \{1, \ldots, n_*\}$,

$$y_\alpha \geq \max_{s \in R_{*\alpha},} Z_{\pi(\beta)} b(v, \pi(\beta))_s + \sum_{s' \in R_{*\beta}} P_{\pi(\beta)}^\infty(s, s') y_\beta$$

$$= \max_{s \in R_{*\alpha},} Z_{\pi(\beta)} b(v, \pi(\beta))_s + y_\beta, \quad \forall\, \alpha = 1, \cdots, n_*.$$

The first term $\max_{s \in R_{*\alpha},} Z_{\pi(\beta)} b(v, \pi(\beta))_s$ is a constant given $v$ and $\pi(\beta)$. Therefore we see that, for any other solution $v + x$ of equation 23, if $y_\beta$ is arbitrarily large then $y_\alpha, \forall\, \alpha = 1, \cdots, n_*$ should also be arbitrarily large. This would violate the Lipschitz assumption on $f$. To see this point, let

$$\tilde{f}(v) \doteq f(\tilde{r} + \tilde{P}v). \tag{32}$$

Let $L$ be the Lipschitz constant of $f$. $L$ is also the Lipschitz constant of $\tilde{f}$ because $\tilde{P}$ is a stochastic matrix and is thus a non-expansion. Fix $v_1 \in \mathcal{V}_\infty$ and choose a $\tilde{v}_2 \in \mathcal{V}_\infty$, denote $m = \|v_1 - \tilde{v}_2\|$. Choose a sufficiently large $d > 0$ such that $\kappa d > L \|v_1 + de - v_2\| = L \|v_1 + de - \tilde{v}_2 - de\| = Lm$, where $v_2 \doteq \tilde{v}_2 + de$. Because $\tilde{f}(v_1 + de) = \tilde{f}(v_1) + \kappa d$, $\left\|\tilde{f}(v_1 + de) - \tilde{f}(v_2)\right\| = \left\|f(v_1) + \kappa d - \tilde{f}(v_2)\right\| = \kappa d > L \|v_1 + de - v_2\|$, which violates the Lipschitz assumption. Because the choice of $\beta$ is arbitrary, $\mathcal{V}_\infty$ is upper bounded.

In addition, because the choice of $\beta$ is arbitrary, we have for any $\alpha \in \{1, \ldots, n_*\}$,

$$y_\alpha \geq \max_{\beta \in \{1, \ldots, n_*\}} \max_{s \in R_{*\alpha},} Z_{\pi(\beta)} b(v, \pi(\beta))_s + y_\beta$$

If $y_\alpha$ is chosen to be arbitrarily small then $y_\beta$ should also be arbitrarily small for all $\beta = 1, \cdots, n_*$ but again this is not allowed due to equation 24 for the same reason. Therefore $y_\alpha, \forall \, \alpha \in \{1, \ldots, n_*\}$ can not be arbitrarily small. Thus $\mathcal{V}_\infty$ is lower bounded. Combining the upper bound and lower bound, $\mathcal{V}_\infty$ is bounded. Therefore $\mathcal{Q}_\infty$ is also bounded.

**Connectedness**

We now show that $\mathcal{Q}_\infty$ is connected. To this end, again it is enough to show that $\mathcal{V}_\infty$ is connected.

Define a function that maps from any $v \in \mathcal{V}$ to a solution in $\mathcal{V}_\infty$. Specifically, fix $c$, let $g : \mathcal{V} \to \mathcal{V}_\infty$ with $g(v) = v + xe$, where $x$ is the solution of $\tilde{f}(v + xe) = c$ and $\tilde{f}$ is defined in equation 32. Note that $x$ is unique given $v$ because $\tilde{f}(v + xe) = \tilde{f}(v) + \kappa x$ and $x = (c - \tilde{f}(v))/\kappa$.

We now show that $g$ is Lipschitz continuous. Consider any $v_1, v_2 \in \mathcal{V}$. Let $x_1, x_2$ satisfy $v_1 + x_1 e = g(v_1)$ and $v_2 + x_2 e = g(v_2)$ respectively. Again $x_1, x_2$ are unique given $v_1, v_2$. Note that

$$\begin{aligned}
&\tilde{f}(v_1 + x_1 e) - \tilde{f}(v_2 + x_1 e) \\
&= \tilde{f}(v_1 + x_1 e) - \tilde{f}(v_2 + x_2 e) + x_1 - x_2 \\
&= c - c + x_1 - x_2 \\
&= x_1 - x_2
\end{aligned}$$

$$\begin{aligned}
|x_1 - x_2| &= |\tilde{f}(v_1 + x_1 e) - \tilde{f}(v_2 + x_1 e)| \\
&\leq L \|v_1 + x_1 e - v_2 - x_1 e\| \\
&= L \|v_1 - v_2\|
\end{aligned}$$

$$\|g(v_1) - g(v_2)\| = \|v_1 + x_1 e - v_2 - x_2 e\| \leq \|v_1 - v_2\| + \|x_1 e - x_2 e\| = (L|\mathcal{S}| + 1) \|v_1 - v_2\|.$$

Therefore $g$ is Lipschitz continuous with Lipschitz constant $L|\mathcal{S}| + 1$.

Finally, because $\mathcal{V}$ is connected and the image of any continuous function on a connected set is connected, $g(\mathcal{V})$ is connected. Note that every point in $g(\mathcal{V})$ belongs to $\mathcal{V}_\infty$ by definition. Every point in $\mathcal{V}_\infty$ is also a point in $g(\mathcal{V})$. To see this point, pick any $x \in \mathcal{V}_\infty$, we can see that $x \in \mathcal{V}$ and that $g(x) = x$. Therefore $\mathcal{V}_\infty = g(\mathcal{V})$ is connected.

Given that $\mathcal{V}_\infty$ is connected, $\mathcal{Q}_\infty$ should also be connected because $\mathcal{Q}_\infty$ is a linear transformation of $\mathcal{V}_\infty$ (see Equation 25).

$\square$

The other result we will need to use to show convergence of the four algorithms introduced in the previous section is the following one. With this result, the stability of the algorithms can be established using the result by Borkar and Meyn (2000) (see also, Section 3.2 by Borkar 2009).

**Lemma 3.** *If an SMDP is weakly-communicating and all rewards are 0, 0 is the only element in $\mathcal{Q}_\infty$.*

*Proof.* Given a weakly-communicating SMDP, by Lemma 1, any solution of the state-value optimality equation equation 23 satisfies

$$v(s) \geq \sum_{s' \in R_*} P_{\pi_*}^\infty(s, s') v(s'), \forall \, s \in R_*, \pi \in \hat{\Pi}_*$$

where $R_*$ is defined right after Lemma 1. Also, because all rewards are 0, $R_*$ is the closed communicating class and $\hat{\Pi}_* = \hat{\Pi}$ contains all stationary policies.

Denote $v_*$ as a solution of equation 23. Now pick an arbitrary policy $\pi \in \hat{\Pi}$. Consider every recurrent class $C$ under $P_\pi$, we have $\forall s \in C$, $v_*(s) \geq \sum_{s' \in C} d_\pi^C(s') v_*(s')$, where $d_\pi^C$ denotes the stationary distribution of $\pi$ in the recurrent class $C$. Sum both sides over $s$ weighted by $d_\pi^C$, $\sum_s d_\pi^C(s) v_*(s) \geq \sum_s d_\pi^C(s) \sum_{s' \in C} d_\pi^C(s') v_*(s') = \sum_{s' \in C} d_\pi^C(s') v_*(s')$. Because l.h.s. = r.h.s. we have $v_*(s) = \sum_{s' \in C} d_\pi^C(s') v_*(s')$. Therefore $v_*(s) = v_*(s'), \forall s, s' \in C$.

Now for any $s, s' \in R_*$, there must exists a $\pi \in \hat{\Pi}_* = \hat{\Pi}$ such that there is a path from $s$ to $s'$ and a path from $s'$ to $s$, because $s, s'$ are in the same communicating class. Therefore $s, s'$ are in the same recurrent class under $P_\pi$. Thus we conclude that $v_*(s) = v_*(s')$. Therefore $\forall s, s' \in R_*$, $v_*(s) = v_*(s')$. The transient states values are uniquely determined by values of states in $R_*$. Thus the solution set of $v$ in the state-value optimality equation equation 23 is also unique up to some additive constant.

The solution set of the option-value optimality equation equation 4 is also unique up to some additive constant. To see this point, let $v(s) = \max_o q(s, o)$, then equation 4 transforms to the state-value optimality equation. Therefore the solution of $q$ in equation 4 satisfies that $\max_o q(\cdot, o)$ is unique up to an additive constant. Let $q_*$ be a solution, $q_*(s, o_1) = q_*(s, o_2), \forall s \in \mathcal{S}, o_1, o_2 \in \mathcal{A}$ because $\forall s, \in \mathcal{S}, o \in \mathcal{O}$:

$$q_*(s, o) = \sum_{s',r} p(s', r \mid s, o) \max_{o'} q_*(s', o')$$
$$= \sum_{s',r} p(s', r \mid s, o) \max_{o'} q_*(s, o')$$
$$= \max_{o'} q_*(s, o').$$

Together with equation 22, the solution of $q$ is uniquely determined. To see that 0 is the unique solution of $q$, we can just verify that 0 is a solution of $q$ in equation 2 and equation 22. The lemma is proved. $\qquad\square$

## 5 PROOF SKETCH OF THEOREM 1-THEOREM 4

In this section, we sketch the proof of Theorem 1-Theorem 4. It has been shown that all four algorithms introduced above are special cases of the *General RVI Q algorithm* (Wan et al. 2021a,b). They also showed that General RVI Q converges under an assumption that is not satisfied for weakly-communicating MDPs/SMDPs. In order to show convergence for weakly-communicating MDPs/SMDPs, we replace this assumption with three weaker assumptions that are satisfied for these MDPs/SMDPs. All other assumptions are the same as those used by Wan et al. (2021a,b) and can be verified for all four algorithms using their arguments. We present General RVI Q and prove its convergence with the three new assumptions in Appendix A.6. The next step of the proof would be verifying the three new assumptions when casting General RVI Q to each of the four algorithms. This should be straightforward given our Theorem 5 and Lemma 3. We delay this part to Appendix A.7. With the three assumptions hold we have the conclusion part of the convergence theorem of General RVI Q holds for each of the four algorithms. The convergence of the reward rates of greedy policies w.r.t. the action/option-values follows the convergence of these values and is shown in Appendix A.8.

## 6 CONCLUSIONS

In this paper, we provide, for the first time, convergence results of off-policy average-reward control algorithms in weakly-communicating MDPs, which are known to be the most general class of MDPs in which it is possible that a learning algorithm can guarantee obtaining an optimal policy. Specifically, we show two existing algorithms, RVI Q-learning and Differential Q-learning, converge in weakly-communicating MDPs. As an extension, we also showed two off-policy average-reward options learning algorithms converge if the SMDP induced by the options is weakly-communicating.

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
