# OpenReview forum: "On Convergence of Average-Reward Off-Policy Control Algorithms in Weakly-Communicating MDPs"
_ICLR.cc/2023/Conference — Submitted to ICLR 2023_

### Official Review · Reviewer_fj9u · 2022-10-23

**Confidence:** 3
**Clarity, Quality, Novelty And Reproducibility:** There are no empirical and demonstrat…
**Correctness:** 3
**Technical Novelty And Significance:** 3
**Empirical Novelty And Significance:** Not applicable
**Recommendation:** 6

**Strength And Weaknesses:**

Strength:
Extension of convergence results under fewer assumptions.

Weakness:
Lack of empirical demonstration of performance gains.

**Summary Of The Paper:**

This paper studies the convergence of off-policy algorithms, which learn a policy that optimizes the average-reward rate using data generated by some other non-controlled policy, for weakly-communicating MDPs, the most general set of MDPs that allow for a learning algorithm that can find under experience a policy of optimal average reward rate. In these MDPs, policies may induce multiple recurrent classes, while some states are transient, while others are reachable from any other state within finite steps. The paper presents results that prove convergence for two learning algorithms, RVI Q-learning and Differential Q-learning, in such MDPs, in a more general setting than has been shown so far.

**Summary Of The Review:**

The achievement of the paper relies on two steps: first, showing that solution sets are non-empty, closed, bounded, and connected; then showing that 0 is the solution to the Bellman equation under rewards 0. Utilizing previous work by Borkar, the convergence results follows. The result goes beyond previous work in that those previous proofs required that all optimal policies induce unichains, which is not always the case in a weakly-communicating MDP. The results are theoretically sound. A discussion on the practical relevance and empirical performance would strengthen the paper.

---

### Official Review · Reviewer_8rbg · 2022-10-24

**Confidence:** 3
**Correctness:** 3
**Technical Novelty And Significance:** 2
**Empirical Novelty And Significance:** 2
**Recommendation:** 3

**Clarity, Quality, Novelty And Reproducibility:**

Overall the paper is well written and easy to read. The related work is also well discussed.

**Strength And Weaknesses:**

Strength: Compared to (Wan et. al., 2021a), the proposed analysis works for general weak-communicating MDPs.

Weakness:
1) The assumptions are too strong to make the theoretical results significant. In particular, in Assumption 4 the authors assume a general coverage ratio for the offline dataset, while in related offline RL papers (e.g., Jin et. al.), it is sufficient to assume a coverage ratio with respect to a optimal policy.

2) The paper is not self-contained. For example, the full proof of Theorem 1 is not presented. I understand that the proof might be very similar to that in (Wan et. al.), but I still suggest the authors to present the full proof or the high-level ideas.

Reference: Is Pessimism Provably Efficient for Offline RL (Jin et. al.)

**Summary Of The Paper:**

The paper studies the performance of differential Q-learning and RVI Q-learning for average-reward weak-communicating MDP.  Main contribution: the authors show that with proper learning rates, the two algorithms converges to the optimal q function (or bias function) assuming that each state-action (state-option) are sufficiently visited.

**Summary Of The Review:**

Currently, the paper does not reach the bar of ICLR in my opinion. My score is 3 and I am willing to increase the score if the author can prove under a weaker assumption. And it would also interesting to find the sufficient and necessary conditions for the trajectory $(S_t,A_t)$ $(t\geq 1)$ to make the two algorithms converge (assume that the learning rates are properly chosen).

---

### Official Review · Reviewer_9oBN · 2022-10-25

**Confidence:** 3
**Correctness:** 3
**Technical Novelty And Significance:** 2
**Empirical Novelty And Significance:** Not applicable
**Recommendation:** 3

**Clarity, Quality, Novelty And Reproducibility:**

Overall the writing and structure of Sections 2-5 are quite muddled and don’t clearly lay out for the reader the structure and story of the exposition.  I give some low-level comments below, but I have two high-level issues:

1) The flow from 3-4-5 is unclear.  I think Section 3 is intended to just be an overview of the results, but this isn’t made explicit.  Then Section 4 starts talking about what sets things converge to and claims this is somehow relevant for the overall convergence, but as far as I can tell nowhere in this section is the relevant ever made clear.  It might help to move 5 before 4 since this at least outlines the overall proof structure and refers to why some of the results in 4 are useful.  Regardless, I think the introductory paragraphs to 3 and 4 need a substantial rewrite to help the reader understand what to expect and why.

2) The handling of the SMDP material is confusing.  I mention one rough transition below, but overall I can’t tell how important this is to the results.  The introduction just describes it as a “direct extension”, but it consumes more than half the preliminaries and the entirety of Section 4 is stated in the language of SMDPs.  Does this play a fundamental role?  Is it just so the more general version is available for the extension and all of 4 would look essentially the same for MDPs?  And how much of this analysis (particularly in 4) is new vs standard?

Small issues:

First paragraph: “The discount factor in the discounted objective has been observed to be deprecated” – needs a cite

Bartlett & Tewari 2009 doesn’t appear in the list of references

The definition of “communicating” is ambiguous regarding the order of quantification.  Does it require there exists a single policy which works for all pairs of states or for all pairs of states such a policy exists?

In the definition of weakly-communicating “closed” is used without definition and I’m assuming the missing word in “that are [] under every policy” is transient.

“shifting ... by any constant vector results in [another] solution”

There is an abrupt transition to “If the agent has a set of options” with no explanation of what an option is or why an agent might have a set of them.  Moving the definition of a SMDP up would help but not entirely solve the rough transition.

Before equation (4), Inter-option Differnential Q-learning needs a cite unless it is actually in Puterman.


**Strength And Weaknesses:**

Describing the strengths of this paper is a bit challenging, in part because I find it lacking in clarity (see below).  There are several things I think I (might) like about the paper, but I am uncertain about how much I like any of them:

1) Extension of prior convergence results to the weakly-communicating setting.  Greater generality is certainly nice from a theoretical perspective, but how much should I care about this result?  The introduction asserts that “it is not rare that an optimal policy introduces multiple recurrent classes”, which would certainly make the extension quite relevant, but the only evidence presented for it seems to be the toy example from Figure 1 and a single citation to Bartlett & Tewari 2009 which is about the generality of this class rather than its importance.  More broadly, the paper is missing any discussion of prior work on weakly-communicating MDPs.

2) Novelty of the technical analysis.  I’m a bit confused because the pitch of the paper is around the weakly communicating setting but Theorem 1 talks about the communicating setting with remark 1 saying weakly-communicating is a minor extension there as well in Theorems 2-4 (which are stated the same way).  From the sketch in 5 and taking a look at the appendix I see that the proof strategy uses the largely the same assumptions as Wan et al. (2021a,b) but replaces their assumption A.8 with new assumptions 9-11.  My guess as to why things are stated the way that they are is that these variant assumptions necessitate some changes to the proof for the communicating case, but as a benefit more easily extend to the weakly-communicating case.  I’m not clear how much a technical delta this represents, as the proofs in the appendix reference the Wan et al. proofs heavily.  I could be convinced (and in fact it seems likely) that there are some non-trivial ideas here, but the paper would need to clearly explain what they are.

This lack of clarity in both the overall exposition and in particular for the articulation of the contribution is the main weakness of the paper.


**Summary Of The Paper:**

This paper examines the convergence properties of existing RL algorithms for the average reward setting when the MDP satisfies a weaker-than-standard assumption on state visitation.  It requires only weak communication, which means that a subset of states is communicating while others may be transient.  For average-reward methods in particular this in introduces technical challenges because the Q-value function is no longer uniquely defined (up to a constant).  The main result is a proof that 4 algorithms known to converge under stronger assumptions also converge in the weakly-communicating setting.

**Summary Of The Review:**

My scores are based on the current draft of the paper with is hard to follow and lacks specificity about the contribution.  I would be much more positively disposed to a substantial rewrite which successfully addresses these issues.

---

### Author Response · Authors · 2022-11-19
**Thanks for your reviews! Here is some additional explanation that might be helpful.**

We would like to thank all reviewers' reviews. First of all, we are glad to see that all three reviewers saw the contribution of the paper,  which is extending the existing convergence result by Wan et al. (2021a;b) from unichain MDPs to weakly communicating MDPs. There are several points that reviewers raised and they would like to see a clearer explanation. And we would like to take this chance to add some additional explanation about them.

1. Reviewer 9oBN pointed out that the paper should have more discussion on the importance of the weakly communicating MDPs, rather than just highlighting the generality of this class of MDPs. And the reviewer would like to see citations to other works that only assumed weakly communicating MDPs. We agree that more citations and better explanations would help readers better understand the importance of this more general class of MDPs. But one important reason for considering weakly communicating MDPs is that this is a quite standard assumption for on-policy average-reward algorithms  (e.g., Jaksch et al. 2010, Ouyang et al. 2017, Zhang & Ji 2019, Wei et al. 2020) but none of the off-policy average-reward algorithms have been shown convergent under this assumption. And our work provides such a result for the first time. We will add more discussion on this point in the revised version of the paper.

2. Reviewer 9oBN also mentioned the technical novelty of the paper is unclear. The main technical novelty is in Section 4. All the stuff there is novel and they are necessary for the extension to communicating and weakly communicating MDPs. Proofs in the appendix are also new but are not technically novel because they are similar to existing results by Wan et al. (2021a;b) and S&F (1978).  The extension from communicating to weakly communicating MDPs is trivial, as discussed in the remark right after Theorem 1. The extension from the unichain case to the communicating case is nontrivial and the entire Section 4 is to show such an extension.

3. Reviewer 9oBN also mentioned the current structure of the paper is not clear because 1) it is not clear why Section 4 plays a key role in the proof of convergence, and 2) transitioning from the MDP case to the SMDP case is abrupt. We agree with the reviewer on this point. In retrospect, we believe that the SMDP case might be better deferred to the appendix. In addition, more explanation about why Section 4 is the key to the proof is indeed helpful for the readability of the paper. Thanks for the reviewer's suggestion.

4. Reviewer 8rbg believed that Assumption 4 is too strong. And the reviewer referred to one paper that uses a weaker assumption. We are completely confused by this comment because our algorithm is online and is for the average-reward formulation, while the algorithm in the referred paper is offline and is for the episodic formulation. Maybe we are wrong but it is not clear to us why one would expect similar assumptions for two algorithms solving two different problems.

5. Reviewer 8rbg also mentioned that the paper is not self-contained because the full proof is missing. This is correct as we only highlighted changes to the proof from Wan et al. (2021a;b). While we do believe that the current technical details are enough for those who have read the proof that this paper is based on, we also understand providing the full proof would provide more convenience to the readers. And we plan to include the full proof in the revision. Thanks for raising this point.

6. Reviewer fj9u is generally positive about our paper. The major concern of the reviewer seems to be the lack of a discussion on the practical relevance and empirical performance. We agree that this point can be improved. Part of the improvement can be moving our empirical result from the appendix to the main text to highlight it.

Ouyang, Y., Gagrani, M., Nayyar, A., & Jain, R. (2017). Learning unknown Markov decision processes: A Thompson sampling approach. Advances in neural information processing systems, 30.

Auer, P., Jaksch, T., & Ortner, R. (2008). Near-optimal regret bounds for reinforcement learning. Advances in neural information processing systems, 21.

Zhang, Z., & Ji, X. (2019). Regret minimization for reinforcement learning by evaluating the optimal bias function. Advances in Neural Information Processing Systems, 32.

Wei, C. Y., Jahromi, M. J., Luo, H., Sharma, H., & Jain, R. (2020, November). Model-free reinforcement learning in infinite-horizon average-reward Markov decision processes. In International conference on machine learning (pp. 10170-10180). PMLR.

---

> ### Comment · Reviewer_9oBN · 2022-12-02
> **Thank you for the explanation**
>
> The explanations regarding my concerns in points 1-3 make a lot of sense and I would be positively disposed toward a future version of this paper which implements these changes.

---

### Decision · Program_Chairs · 2023-01-20

**Decision:**

Reject

**Justification For Why Not Higher Score:**

There are concerns about the strong assumptions and novelties.

**Justification For Why Not Lower Score:**

N/A

**Metareview: Summary, Strengths And Weaknesses:**

This paper established new convergence results for weakly communicating MDPS. Reviewers raised concerns about the assumptions and novelties. The AC agrees and recommends rejection.